# SmoQyDEAC.jl: A differential evolution package for the analytic continuation of imaginary time correlation functions

James Neuhaus[1,2] , Nathan S. Nichols[3] , Debshikha Banerjee[1,2] ,
Benjamin Cohen-Stead[1,2] , Thomas A. Maier[5] , Adrian Del Maestro[1,2,4] , and
Steven Johnston[1,2,*]

**1** Department of Physics and Astronomy, The University of Tennessee, Knoxville, TN 37996,
USA
**2** Institute of Advanced Materials and Manufacturing, The University of Tennessee, Knoxville,
TN 37996, USA
**3** Argonne Leadership Computing Facility, Argonne National Laboratory, Argonne, Illinois
60439, USA
**4** Min H. Kao Department of Electrical Engineering and Computer Science, University of
Tennessee, Knoxville, Tennessee 37996, USA
**5** Computational Sciences and Engineering Division, Oak Ridge National Laboratory, Oak
Ridge, Tennessee 37831-6494, USA

⋆ sjohn145@utk.edu

October 1, 2024

## Abstract

We introduce the `SmoQyDEAC.jl` package, a Julia implementation of the Differential Evolution Analytic Continuation (DEAC) algorithm [**N. S. Nichols** *et al.*, **Phys. Rev. E 106, 025312 (2022)**] for analytically continuing noisy imaginary time correlation functions to the real frequency axis. Our implementation supports fermionic and bosonic correlation functions on either the imaginary time or Matsubara frequency axes, and treatment of the covariance error in the input data. This paper presents an overview of the DEAC algorithm and the features implemented in the `SmoQyDEAC.jl` package. It also provides detailed benchmarks of the package's output against the popular maximum entropy and stochastic analytic continuation methods. The code for this package can be downloaded from our GitHub repository at https://github.com/SmoQySuite/SmoQyDEAC.jl or installed using the Julia package manager. The online documentation, including examples, can be accessed at https://smoqysuite.github.io/SmoQyDEAC.jl/stable/.

# 1 Introduction

## 1.1 Background and Motivation

Theoretical approaches for solving quantum many-body Hamiltonians are often developed using the Matsubara formalism, where correlation functions are calculated in imaginary time $\tau$ or Matsubara frequency $i\omega_n$. However, to obtain results relevant to experiments, one must analytically continue the imaginary-time results to the real-frequency axis. For example, a common problem in many-body theory is to obtain the spectral function $\mathcal{A}(\omega)$ from a Matsubara frequency correlation function $\mathcal{G}(i\omega_n)$, where $\omega_n = \pi(2n+1)/\beta$ for fermions or $2n\pi/\beta$ for bosons, $n \in \mathbb{Z}$ is an integer, and $\beta$ is the inverse temperature. These quantities are related by the integral equation

$$\mathcal{G}(i\omega_n) = \int_{-\infty}^{\infty} d\omega \, \frac{\mathcal{A}(\omega)}{i\omega_n - \omega}, \tag{1}$$

where we set $\hbar = k_B = 1$. Alternatively, the corresponding imaginary time correlation function $\mathcal{G}(\tau)$, the Fourier-transform of $G(i\omega_n)$, is related to the spectral function by the integral equation

$$\mathcal{G}(\tau) = \int_{-\infty}^{\infty} d\omega \, \mathcal{A}(\omega) \frac{e^{-\tau\omega}}{1 \pm e^{-\beta\omega}}, \tag{2}$$

where the $+$ $(-)$ sign is for fermions (bosons). Here, we use the convention that correlation functions are strictly positive on the interval $\tau \in [0, \beta)$ and defined as

$$\mathcal{G}_{A,B}(\tau) = \langle A(\tau)B(0) \rangle \tag{3}$$

for arbitary operators $A$ and $B$.

If the spectral function is known, or in cases where the Matsubara correlation function can be expressed as a sum of simple poles, then the real-axis retarded correlation function can be

obtained by performing a simple Wick rotation $i\omega_n \rightarrow \omega + i\delta$, where $\delta = 0^+$ is an infinitesimal positive number. But for many problems of interest, the imaginary time correlation functions are calculated using numerical methods, and Eqs. (1) or (2) must be inverted to obtain the spectral function. Inverting these equations is challenging, however, because the kernels of Eqs. (1) or (2) are ill-conditioned such that the resulting solutions for the spectral function are no longer unique and can be very sensitive to noise in the correlation function data.

Some theoretical approaches, such as diagrammatic perturbation theory, enable noiseless calculations of correlation functions within machine precision. In these cases, one can use techniques like Padé approximants [1] or Nevanlinna analytic continuation (AC) [2] to obtain the spectral functions numerically. One can also sometimes use mixed representations of the correlation functions to get numerically stable and unique solutions for the retarded correlation functions $\mathcal{G}(\omega + i\delta)$ [3, 4]. These methods, however, are not well suited for performing numerical analytic continuation if the correlation functions are inherently noisy.

Quantum Monte Carlo (QMC) methods represent a powerful class of numerical techniques for solving quantum many-body systems [5, 6]. Typically, QMC algorithms work by mapping a $D$-dimensional interacting quantum problem onto an equivalent $(D + 1)$-dimensional non-interacting problem by introducing auxiliary fields that decouple the interactions. These fields are then integrated out using an appropriate sampling method (e.g. Metropolis-Hastings [7], Langevin [8, 9], or hybrid Monte Carlo [10–12]), where QMC performs a weighted "random walk" through configuration space and periodically measures the relevant correlation functions. The correct interacting result for the original quantum system is recovered by averaging over many such measurements. The non-deterministic nature of QMC necessitates generating a large volume of sample data but results in robust, though noisy, estimates for the measured observables in imaginary time.

As mentioned above, performing numerical analytic continuation on noisy correlation functions is challenging due to the ill-conditioned kernels in Eqs. (1) and (2). An early approach developed to tackle this problem, which has been widely adopted, is the maximum entropy method (MEM) [13]. It utilizes Bayesian approaches to infer $\mathcal{A}(\omega)$ by tuning an internal hyper-parameter $\alpha$ to generate a maximally likely distribution. Here, $\alpha$ controls a balance between fitting the measured correlation function data and deviations from a user-specified "default spectrum," which encodes known (or assumed) information about the true spectral function. Multiple methods exist to find $\alpha$, with the Bryan algorithm [14] and the newer $\chi^2$ kink [15] methods being used widely. Two popular alternatives to MEM are stochastic analytic continuation (SAC) [16] and the stochastic optimization method (SOM) [17]. As the names imply, both algorithms are stochastic techniques that utilize Metropolis sampling [7] to find multiple prospective spectral functions, which are then averaged to obtained the final predicted spectrum.[1]

Recently, Nichols *et al.* [19] have introduced another stochastic approach to this problem, the differential evolution for analytic continuation (DEAC) algorithm. DEAC optimizes the spectral function by evolving and mutating a population of candidate spectra (genomes) to achieve an ideal fit of the measured correlation functions, providing a robust update of the previously introduced genetic inversion via falsification of theories approach [20]. Each genome attempts to improve its $\chi^2$ goodness of fit to the measured correlation function by proposing cross mutations within their genes and updating the genome when a better fit is obtained. Like SOM and SAC, the DEAC algorithm provides a means to obtain the spectral function without making prior assumptions about the spectra itself. Crucially, DEAC is often able to produce results comparable to or better than SOM or MEM while requiring fewer computational resources [19].

The DEAC algorithm was originally formulated for and applied to analytically continuing

---

[1]For a recent review of these methods, we direct the reader to Ref. [18].

bosonic correlation functions [19]. In this paper, we introduce the `SmoQyDEAC.jl` package, a user friendly implementation of the DEAC algorithm and part of the `SmoQySuite` of codes [21] that implements the DEAC algorithm with extended support for both boson and fermion correlation functions.

## 1.2 Scope & Overview

This paper introduces the `SmoQyDEAC.jl` package and is intended to serve as a citable document for when the package is used in research. It is not intended to serve as a detailed user manual. Instead, we refer users to our online documentation, which provides extensive examples and scripts.

The remainder of this document is organized as follows. Section 2 describes the algorithm [19] at `SmoQyDEAC.jl`'s core. Section 3 provides an overview of the package, instructions for installation, a brief overview of the API, the package's multi-threading support, and other features. Section 4 then presents multiple test cases, where we benchmark `SmoQyDEAC.jl` against other popular AC methods using synthetic fermionic and bosonic correlation functions. That section also compares results for the single-particle electron and phonon spectral functions of the half-filled one-dimensional (1D) Hubbard-Holstein and Holstein models obtained from density matrix renormalization group (DMRG) and determinant quantum Monte Carlo (DQMC) simulations, where the latter are analytically continued using DEAC. Our results demonstrate that our implementation of the DEAC algorithm can achieve results comparable to or better than alternative methods and can reproduce sharp and subtle features in the spectral functions such as dispersion kinks.

## 2 DEAC Algorithm

The DEAC algorithm is outlined in Algorithm 1 [19]. It is a genetic algorithm-based approach to the inverse problem of extracting a spectral function $\mathcal{A}(\omega)$ from noisy, discretized QMC correlation function data on the imaginary time $[\mathcal{G}(\tau)]$ or Matsubara frequency $[\mathcal{G}(i\omega_n)]$ axis [19]. For brevity we will only describe the algorithm for imaginary time, though the package also supports Matsubara frequency input. We will further assume that the correlation function $\mathcal{G}(\tau)$ has been measured on a discrete imaginary time grid $\tau_l = l\Delta\tau$, where $\Delta\tau = \beta/N_\tau$ and $l = 0, 1, \ldots, N_\tau - 1$. We further denote the error of the measurement of $\mathcal{G}(\tau = l\Delta\tau)$ as $\sigma_l$. Similarly, the spectral function $A(\omega)$ is defined on a fixed frequency grid $\omega_j = -\omega_c + j\Delta\omega$, where $\omega_c$ is a frequency cut-off, $N_\omega$ is the number of frequency points, $\Delta\omega = 2\omega_c/N_\omega$ is the frequency spacing, and $j = 0, 1, \ldots, N_\omega - 1$. The values of the spectral function at each frequency act as the genes in the DEAC algorithm, and the array of values specifying the spectral function is the genome, which is updated following the evolutionary process described below. In the following discussion, we will focus on the evolution of a single population of genomes for simplicity. In practice, however, we execute many parallel processes and average the results to improve statistics and obtain a smoother final spectrum. Details of `SmoQyDEAC.jl`'s options for parallelization are discussed in Sec. 3.3.

DEAC finds an optimal spectral function $\mathcal{A}(\omega)$ by evolving (optimizing) a population of trial genomes $\mathcal{A}_i(\omega)$, $i \in \{1, ..., N_{\text{pop}}\}$, that will cross-mutate in an attempt to improve a fitness function

$$\chi_i^2 = \frac{1}{N_\tau} \sum_{l=0}^{N_\tau-1} \frac{[\mathcal{G}(\tau_l) - \mathcal{G}_i(\tau_l)]^2}{\sigma_l^2}, \tag{4}$$

**Algorithm 1** Differential Evolution for Analytic Continuation

> $\forall i : \gamma_i = \gamma_{\text{default}}$.
> $\forall i : \delta_i = \delta_{\text{default}}$.
> Randomly initialize all $\mathcal{A}_i(\omega)$.
> $\forall i$: Calculate $\chi_i^2$ using Eqs. (2) and (4)
> **for** generation $\in \{1, \ldots, N_{\text{generations}}\}$ **do**
>     **for** $i \in \{1, \ldots, N_{\text{genomes}}\}$ **do**
>         Randomly update $\gamma_i \in [0, 1)$ and $\delta_i \in [0, 2)$
>         Sample three unique trial genomes $\{j, k, l\}$
>         Randomly assign which portion of genes $\propto \gamma_i$ to update
>         Generate proposed trial genome $\mathcal{A}_i'(\omega)$ according to Eq. (7)
>         Calculate $\chi_{i,\text{new}}^2$ for proposed updates
>         **if** $\chi_{i,\text{new}}^2 < \chi_i^2$ **then**
>             $\mathcal{A}_i(\omega) \leftarrow \mathcal{A}_i'(\omega)$
>             $\chi_i^2 \leftarrow \chi_{i,\text{new}}^2$
>         **end if**
>     **end for**
>     **if** any $\chi_i^2 < \chi_{\text{threshold}}^2$ **then**
>         **break** generation loop
>     **end if**
> **end for**
> $i_{\text{best}} \leftarrow \arg\min\left[\chi_i^2\right]$
> $\mathcal{A}(\omega) \leftarrow \mathcal{A}_{i_{\text{best}}}(\omega)$

where

$$\mathcal{G}_i(\tau_l) = \sum_{j=0}^{N_\omega - 1} \Delta\omega K\left(\tau_l, \omega_j\right) \mathcal{A}_i\left(\omega_j\right) \tag{5}$$

and $K(\tau, \omega)$ is a kernel function whose form depends on the type of correlation function

$$K(\tau, \omega) = \begin{cases} \frac{e^{-\omega\tau}}{1 + e^{-\beta\omega}} & \text{Fermionic} \\ \frac{e^{-\omega\tau}}{1 - e^{-\beta\omega}} & \text{Bosonic} \\ \frac{e^{-\omega\tau} + e^{-(\beta-\tau)\omega}}{1 - e^{-\beta\omega}} & \text{Symmetrized Bosonic} \end{cases}. \tag{6}$$

Note, $\chi_i^2 \equiv \chi_i^2[\mathcal{A}_i(\omega)]$ is a functional of the trial genome $\mathcal{A}_i(\omega)$ but we will drop this notation unless it is needed for clarity.

The DEAC algorithm uses two internal parameters to control the generation of data: the trial genome crossover probability $\gamma_i$ and trial genome differential weight $\delta_i$. The trial genome crossover probability $\gamma_i \in [0, 1)$ determines the probability of any gene being updated in each generation while trial genome differential weight $\delta_i \in [0, 2)$ determines the ratio of gene mixing. Specifically, the $i^{\text{th}}$ trial genome is replaced by the mixing of three parent genomes $(j, k, l)$ selected from the same population according to the rule

$$\mathcal{A}_i'(\omega) = \left|\mathcal{A}_j(\omega) + \delta_i\left[\mathcal{A}_k(\omega) - \mathcal{A}_l(\omega)\right]\right|. \tag{7}$$

(This rule is shown graphically in Fig. 1.) For each $i$, the values of $(j, k, l)$ are randomly drawn from the population without replacement such that all four indices are unique. Each update is accepted $\mathcal{A}_i'(\omega) \rightarrow \mathcal{A}_i(\omega)$ if it reduces the overall fitness

$$\chi_i^2[\mathcal{A}'(\omega)] < \chi_i^2[\mathcal{A}(\omega)] \tag{8}$$

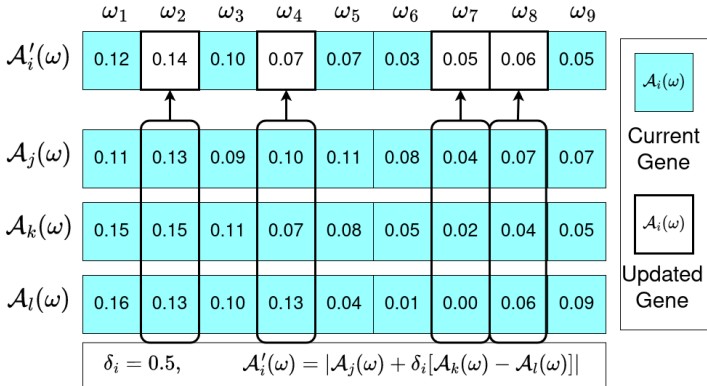

Figure 1: Example proposed genome update for trial genome $\mathcal{A}_i(\omega)$, assuming a differential weight $\delta_i = 0.5$. Boxed genes are those randomly selected to update using the $\gamma_i$ parameter.

and rejected otherwise. In other words, only spectra with improved $\chi^2$ fits are updated.

This process then repeats until either a maximum number of generations has been reached or until any of the trial genomes meets a fitness threshold

$$\exists i \in N_{\text{pop}} : \chi_i^2 < \chi_{\text{threshold}}^2. \tag{9}$$

The estimate for the spectral function is then selected as the genome with the best fit

$$i_{\text{best}} = \arg\min\left[\chi_i^2\right], \tag{10}$$

such that $\mathcal{A}(\omega) = \mathcal{A}_{i_{\text{best}}}(\omega)$ is the final reported spectral function.

## 3 The SmoQyDEAC.jl package

### 3.1 Package Information

SmoQyDEAC.jl is a standalone package in the SmoQySuite organization [21]. It implements the DEAC algorithm using a streamlined application programming interface (API) to conduct AC on correlation functions from finite temperature QMC simulations. The SmoQyDEAC.jl package adopts a design philosophy similar to other packages in the SmoQySuite organization in that it prioritizes ease of installation and use while remaining efficient.

We chose to implement SmoQyDEAC.jl in the Julia programming language [22] for several reasons. First, the Julia language includes an integrated package manager, ensuring that SmoQyDEAC.jl is easy to install, with all dependencies automatically installed and linked without user intervention. Second, Julia combines the benefits of compiled and interpreted languages. Just-in-time compilation allows Julia to match the performance of compiled languages that are frequently used for intensive scientific computations, such as FORTRAN and C/C++. At the same time, scripting functionality renders Julia similarly convenient for post-processing and analyzing data as languages like Python and Matlab, including the use of Jupyter notebooks [23].

The code for SmoQyDEAC.jl can be downloaded from our GitHub repository at https://github.com/SmoQySuite/SmoQyDEAC.jl, or the package can be installed using the Julia package manager by issuing the following commands:

```
julia> ]
pkg> add SmoQyDEAC
```

The corresponding online documentation, including examples, can be accessed at https://smoqysuite.github.io/SmoQyDEAC.jl/stable/.

## 3.2 API

SmoQyDEAC.jl has a simplified API with two outward facing functions, DEAC_Binned(...) and DEAC_Std(...), which have support for correlation functions measured on the $\tau$ or $i\omega_n$ axis. DEAC_Binned(...) takes as input QMC binned data from which it calculates and diagonalizes the covariance matrix. It then rotates the average correlation functions and kernel functions $[K(\tau, \omega)$ or $K(\omega_n, \omega)]$ into the eigenbasis of the covariance matrix, and performs the $\chi^2$ fitting in this basis.

DEAC_Std(...) utilizes the final averaged correlation function data and corresponding standard error to fit the spectral functions. While DEAC_Std(...) is given as an option for the user, it is not recommended if binned data is available. Time-displaced correlation functions generated by finite temperature QMC simulations have correlated noise in imaginary time, and whenever possible, these correlations should be accounted for by using DEAC_Binned(...) [13,24]. Ideally, the number of bins should be greater than the number of imaginary timesteps $N_\tau$. Therefore, SomQyDEAC.jl uses bootstrap resampling [25] to ensure that at least $5N_\tau$ or $5N_{\omega_n}$ "bins" are used to calculate the covariance matrix.

When utilizing the covariance method, some values of the input correlation functions may not be linearly independent. For example, many bosonic correlation functions are symmetric about $\beta/2$ while diagonal elements of fermionic Green's functions must satisfy $G(0)+G(\beta) = 1$. Thus, some of the covariance matrix eigenvalues may be zero (within machine precision). To address this issue, any eigenvalue that has a ratio against the largest eigenvalue below a cutoff parameter eigenvalue_ratio_min will not be used for $\chi^2$ fitting.

The final spectral functions provided by SmoQyDEAC.jl are reported on evenly spaced $\omega$ grids. If the output is for a symmetric bosonic function, then only the data at positive $\omega$ values are reported.

Example scripts using the SmoQyDEAC.jl API and correct input formatting can be found in the online documentation.

## 3.3 Parallelization

A single run of the DEAC algorithm generally produces a noisy spectrum for $\mathcal{A}(\omega)$. Therefore, it often is necessary to run the algorithm $\mathcal{O}(10^3)$ times, each using a unique random seed, to reconstruct reasonably smooth spectral functions. SmoQyDEAC.jl includes built-in multi-threading support to facilitate this process. Each independent run is assigned its own thread using Julia's native thread manager. When enough runs have reached their final fitness, the package will bin the data and generate a checkpoint file. To take advantage of multithreading, you must run Julia with threads enabled by invoking the threads command line switch with either the number of threads, e.g.,

```
$ julia --threads=4 [YourScript].jl
```

or auto, e.g.,

```
$ julia --threads=auto [YourScript].jl
```

Using auto will tell Julia to find and use the available number of logical cores on the computer or node on which it is run. Because of the stochastic nature of the DEAC algorithm, one might

be tempted to associate the statistics for each spectral function with confidence bands on the final spectrum. However, as in SAC, such error estimates often don't reflect the quality of fit [18]. Our results for a bosonic test case shown in Fig. 4, where the corresponding confidence bands are shown in light blue, illustrate this limitation.

It is important to note that each iteration of the DEAC algorithm can obtain different overall fitness measures, and each is not guaranteed to converge to the desired fitness within a maximum number of generations. Therefore, the total number of runs performed is split into equal-sized bins. Then, to account for varying fitness levels, `SmoQyDEAC.jl` performs a weighted average of the runs in each bin with a weight proportional to the square of the inverse fitness

$$\mathcal{A}_{\text{bin}}(\omega) = \frac{\sum_\mu \chi_\mu^{-2} \mathcal{A}_\mu(\omega)}{\sum_\mu \chi_\mu^{-2}}, \tag{11}$$

where $\mathcal{A}_\mu(\omega)$ and $\chi_\mu^2$ are the spectral function and fitness for run $\mu \in [1, N_{\text{b}}]$ in the current bin and $N_{\text{b}}$ is the total number of runs per bin. The final returned spectral function is given by the average of the $\mathcal{A}_{\text{bin}}(\omega)$ spectral functions associated with each bin. If you find that multiple runs are having trouble reaching a target $\chi^2$, we recommend increasing the `population_size` parameter or the `number_of_generations` parameter in your DEAC function call.

Appendix A provides an example of the expected reduction in the noise for the final predicted spectrum as the total number of genomes or bins is increased.

## 3.4 Fitness Floor Finder

The $\chi^2$ fits found by `SmoQyDEAC.jl` are normalized by the number of degrees of freedom in an attempt to give an ideal converged fitness of $\sim 1$. However, a user may wish to explore fits below this number. For $\chi^2 < 1$, DEAC's change in fitness per generation will suffer from diminishing returns after many generations. To prevent exceedingly long run times and the need for a large number of maximum generations, we have provided a fitness floor finder (FFF) utility, which will try to detect when further gains in fitness are unlikely to occur. Before calculating bins of $\mathcal{A}_i(\omega)$, the FFF will run the DEAC algorithm for a minimum of ten runs and up to the number of available threads. If a thread sees $\chi_{\alpha,\text{best}}^2$ improve less than 10% for over two consecutive 10,000 generation spans, it will halt and store its best current $\chi_{\alpha,\text{best}}^2$ value. The FFF then sets $1.025 \times \min\left[\chi_\alpha^2\right]$ as the lowest value for $\chi_{\text{threshold}}^2$. FFF testing results can be found in Appendix B.

To quantify goodness-of-fit, one must be careful about delving below the generally accepted value of $\chi^2 = 1$. As in SAC, lowering $\chi^2$ targets below 1 slightly improves performance in some tests [18]. However, this risks overfitting noise, resulting in an unrealistic spectrum. For a more exhaustive discussion on lowering your target $\chi^2$ while avoiding overfitting, see Ref. [18] or Appendix C.

## 3.5 User Defined Mutation

`SmoQyDEAC.jl` also provides functionality for a users to introduce their own mutation rules by passing a user defined function to the `user_mutation!` argument. The DEAC algorithm then proposes and applies updates based on the user defined mutation serially and independently within a generation. An example implementation, as well as further details of this functionality, are provided in the `SmoQyDEAC.jl` documentation.

## 4   Testing and Benchmarks

### 4.1   Generating Synthetic Correlation Function Data

To test the `SmoQyDEAC.jl` package and benchmark it against other popular AC methods, we generated several synthetic data sets. Because we are interested in analytically continuing correlation functions generated by Markov chain QMC simulations, care must be taken to properly mimic the correlated noise inherent in these simulations. Naively modeling measurement uncertainty by introducing statistically independent random noise at each imaginary time is a poor reflection of real data, where the noise in time-displaced Green's function measurements is typically correlated to some degree in imaginary time. In light of this, we utilize the method by Shao *et al.* [24] to generate noisy, correlated synthetic data for our testing. First, we define a known spectrum $\mathcal{A}_{\text{true}}(\omega)$, which is then used to generate a corresponding $G_{\text{true}}(\tau_i)$ via the transformation

$$G_{\text{true}}(\tau_i) = \sum_{j=0}^{N_\omega - 1} K(\tau_i, \omega_j) \mathcal{A}_{\text{true}}(\omega_j) \Delta\omega, \tag{12}$$

with the relevant kernal $K(\tau, \omega)$ as given in Eq. (6). Next, we generate a set of normal-distributed random numbers $\sigma_j^0$ with an initial standard deviation $\sigma^0$. Then, using an auto-correlation parameter $\xi$, we introduce correlated noise in imaginary time using the formula

$$G_{\text{bin}}(\tau_i) = G_{\text{true}}(\tau_i) + \frac{\sum_j \sigma_j^0 e^{-|\tau_i - \tau_j|/\xi}}{\sqrt{\sum_j e^{-2|\tau_i - \tau_j|/\xi}}}. \tag{13}$$

All of the imaginary time correlation functions used in our testing are strictly positive for $\tau \in [0, \beta)$. To enforce this condition, we randomly sample noisy values for the Green's function from a Gamma distribution with mean $G_{\text{true}}(\tau_i)$ and standard deviation $\sigma^0$ for each imaginary time $\tau_i$. We then subtract off the true value $G_{\text{true}}(\tau_i)$ to calculate the $\sigma_j^0$ values used in Eq. (13) above. This ensures $G_{\text{bin}}(\tau_i) \geq 0$ for both bosonic and fermionic test cases. The Julia code used to generate our test cases is publicly available at https://github.com/sandimas/SynthAC.jl.

### 4.2   Alternative Analytic Continuation Methods for Comparison

In what follows, we will compare the output of the DEAC algorithm with results generated using the SAC method and two different implementations of the MEM method. These methods are widely used in the literature and comparing the results of all four algorithms provides useful information about the relative strengths of each approach.

SAC parameterizes a spectral function as a series of $\delta$-functions that are stochastically updated via Monte Carlo sampling [7, 16] to minimize a distance metric between proposed and measured correlation functions. For our test cases, we utilized Jeff Wang's C++ implementation of SAC [26], which was run using the provided default settings.

The MEM method is the most commonly used framework for solving the analytic continuation problem and is based on Bayesian inference. It regularizes the inverse problem through the introduction of an entropy-like term that measures the deviation of the predicted spectrum from a default spectrum and then determines the most probable spectrum $A(\omega)$ using deterministic optimization. Here, we compared our results against two different implementations of the MEM algorithm. The first was the MEM capabilities provided by the ACFlow package [27]. Currently, ACFlow does not support utilizing the full covariance matrix, but it does include the $\chi^2$ kink method for hyper-parameter tuning [15]. We additionally use our own implementation of the MEM algorithm that utilizes the covariance matrix and the Bryan Algorithm for hyper-parameter tuning [14]. For all MEM runs, we adopted Gaussian default

model centered at $\omega = 0$ with a standard deviation of $\sigma = 4$ for the fermionic tests and a flat default model for the bosonic ones. Both MEM codes include an additional factor of $\omega$ in their bosonic kernels, so the default model does not correspond to the spectral function $A(\omega)$ in these cases but rather $A(\omega)/\omega$.

For our DEAC reference, we ran `SmoQyDEAC.jl` with the default settings with a maximum of 100,000 generations per genome and a population of 4,000 total genomes (spectral functions).

## 4.3 Results for Fermionic Correlation Functions

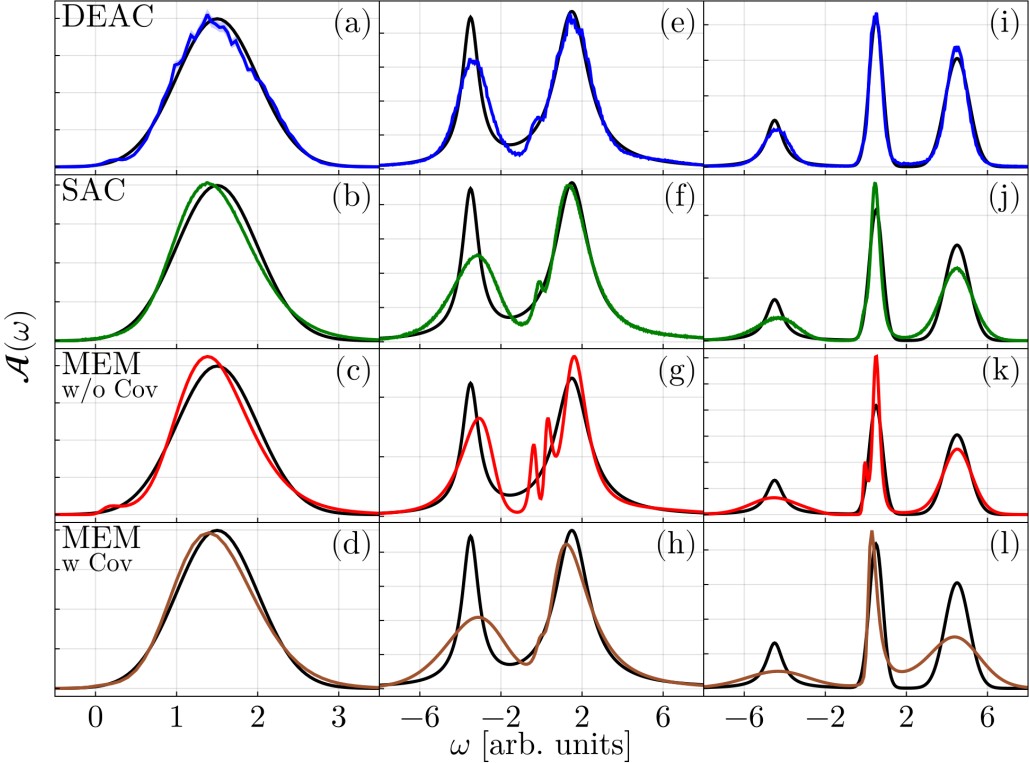

Figure 2: Representative test results for fermionic correlation functions. The left column, panels (a)-(d), shows results for $\mathcal{A}_{\text{true}}(\omega)$ consisting of a single a Gaussian centered at $\mu = 1.5$, width $\sigma = 0.5$. The middle column, panels (e)-(h), show results for $\mathcal{A}_{\text{true}}(\omega)$ given by a sum of two weighted Lorentzian distributions centered at $\mu = -3.5$ and $\mu_2 = 1.5$ with half-width at half-maximum (HWHM) values of $\Gamma_1 = 0.5$ and $\Gamma_2 = 1$, respectively. The right column, panels (i)-(l), show results for $\mathcal{A}_{\text{true}}(\omega)$ given by a sum of a single Lorentzian distribution ($\mu = -4.5$, $\Gamma = 0.5$) and two Gaussian distributions ($\mu_1 = 0.5$, $\sigma_1 = 0.33$ and $\mu_2 = 4.5$, $\sigma_2 = 0.6$). In all panels, a black line represents the ground truth the the colored line is the reconstruction.

We chose a series of increasingly difficult test cases ranging from a single Gaussian to complicated multi-peak spectra with a mix of broad and sharp peaks that are often difficult to reproduce using standard AC methods. We will discuss a subset of particular informative examples here, with additional test cases available at https://zenodo.org/records/10407525. In what follows, we will focus the discussion on the qualitative aspects of the results, as quantitative metrics may not always account for feature replication.

Our first test case, shown in Figs. 2(a)-(d), was a single Gaussian centered at $\omega = 1.5$ and

with a standard deviation of $\sigma = 0.5$. All four tested methods could reproduce the spectral function well in this case.

Our second test case, shown in Figs. 2(e)-(h), consisted of the sum of two Lorentz (Cauchy) distributions of differing means, widths, and amplitudes. (The exact values of these parameters are provided in the figure caption.) All four methods find the correct location and magnitude for the second broader peak, though MEM without the full covariance matrix produces a spectrum that is oscillatory along its ascending slope. While all four methods also capture the low-energy peak, they generally over predict its width and under-predict its height with DEAC performing slightly better in this regard.

Our third test case, shown in Figs. 2(i)-(l), consisted of a single Lorentzian peak at $\omega = -4.5$ summed with two Gaussians. Here, both MEM methods and SAC overpredict the width of the Lorentzian peak significantly and underpredict the third peak's magnitude. In this case, DEAC performs best; it slightly underfits the first peak and replicates both Gaussian peaks at higher energy very well. Overall, we find that DEAC is compatible to MEM and SAC when applied to a single broad peak but outperforms these methods when applied to multi-peaked fermionic spectral functions.

## 4.4 Results for Bosonic Correlation Functions

We also tested `SmoQyDEAC.jl` with a series of bosonic correlation functions and a symmetric kernel [see Eq. (6)]. Bosonic spectral functions are odd functions with respect to energy such that $\mathcal{A}(-\omega) = -\mathcal{A}(\omega)$. To enforce this, we first generated test distributions $s(\omega)$, which were then reflected across the $\omega = 0$ axis and multiplied by a factor of $\omega$ such that

$$\mathcal{A}(\omega) = \omega\left[s(\omega) + s(-\omega)\right]. \tag{14}$$

Similar to the fermion tests, our first case, shown in Figs. 3(a)-(d), consists of a single Gaussian. Here, the two MEM methods provide the best results, while the SAC result skews the peak slightly to lower energies and overpredicts the total weight while DEAC artificially narrows the distribution.

Our second test case, shown in Figs. 3(e)-(h), is a Lorentzian distribution, which is made asymmetric at low energies due to the odd symmetry condition imposed by Eq. (14). In this case, MEM without the co-variance matrix reproduces the true spectral function while incorporating the co-variance slightly broadens the peak with the predicted spectra weight shifted a little toward to higher energies. The SAC method reproduces the general shape of the spectrum but significantly overestimates its magnitude. Finally, the DEAC algorithm also reproduces the true spectrum well but with quite a bit of noise, which could be reduced with better statistics.

Our third test case, shown in Figs. 3(i)-(l), consists of two Gaussian lineshapes with different widths. Here, DEAC and MEM without covariance reproduce the leftmost peak well while narrowing the broader Gaussian at higher energy. The SAC captures the position of both peaks but over-estimates the magnitude of the spectral function at all energies. Finally, the MEM with covariance produces a spectrum where the weight of both peaks is shifted to higher energies.

We have often observed that DEAC has difficulty capturing bosonic spectral functions with a sharp peak centered at low energies $\omega$ when the symmetric bosonic kernel is used. Fig. 4(a), investigates this issue in more detail by comparing results for a particularly challenging case — a narrow Guassian distribution centered at $\mu = 0.1$. Here, the SAC and MEM methods converge to similar solutions with a broadened peak, while DEAC gives a shark-tooth-like lineshape. While DEAC replicates the tail at higher $\omega$, the peak is shifted significantly. Shifting the peak to an even smaller $\omega$ value, as shown in Fig. 4(b), DEAC produces a spectrum with spectral weight piled up at zero energy with the peak's maximum at roughly half the true $\omega$.

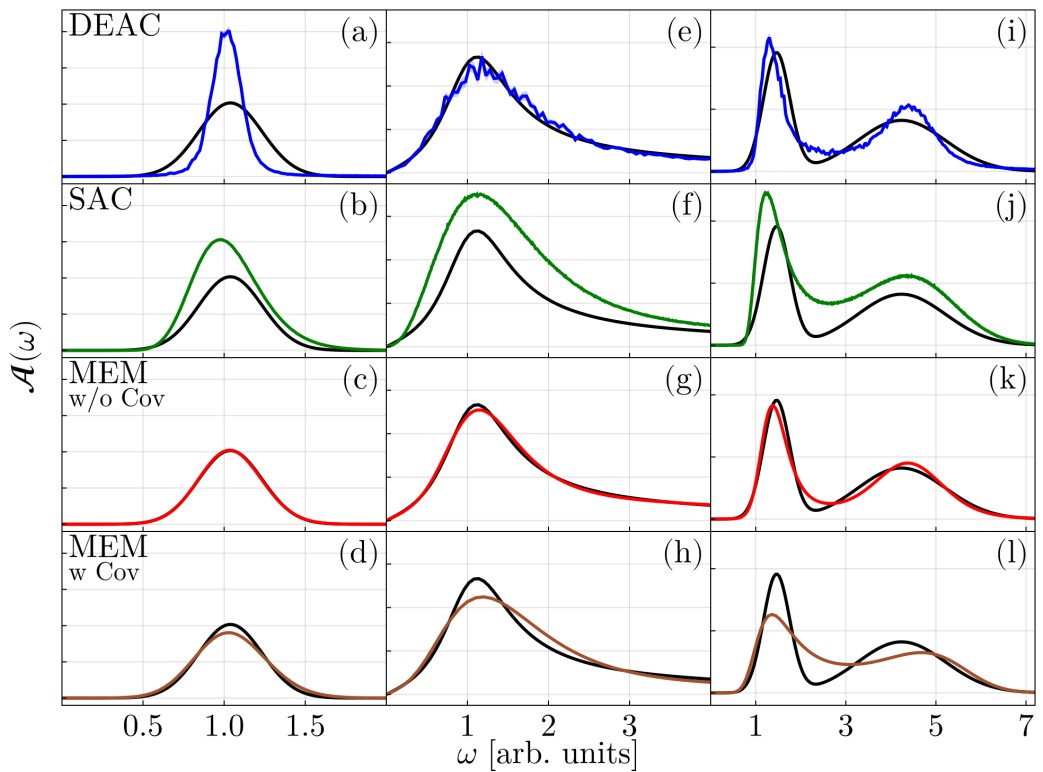

Figure 3: Three bosonic comparisons of AC methods. (a)-(d) $s(\omega)$ is a Gaussian centered at $\mu = 1.0$, width $\sigma = 0.2$. (e)-(h) $s(\omega)$ is a Lorentzian $\mu = 1.0$, width $\sigma = 0.5$. (i)-(l) $s(\omega)$ is a superposition of two weighted Gaussians with $\mu = 1.4$, $\sigma = 0.3$ and $\mu = 4.0$, $\sigma = 1.0$

Meanwhile, SAC and MEM methods are able to replicate the peak reasonably well, with the ACFlow MEM and the $\chi^2$ kink method giving nearly perfect results.

## 4.5 Application to the 1D Holstein model

For our final set of tests, we applied `SmoQyDEAC.jl` and the MEM methods to data generated in low-temperature DQMC simulations [28] of the half-filled Holstein and Hubbard-Holstein chains. By focusing on a 1D system, we can obtain results with good momentum resolution and compare them to DMRG calculations [29] performed directly on the real frequency axis. All results in this section were obtained on $L = 32$ chains.

The Hamiltonian for the 1D Holstein model is

$$\hat{H} = -t \sum_{i,\sigma} \left( \hat{c}^\dagger_{i,\sigma} \hat{c}_{i+1,\sigma} + \text{h.c.} \right) - \mu \sum_{i,\sigma} \hat{n}_{i,\sigma} + \sum_i \left( \frac{1}{2M} \hat{P}_i^2 + \frac{1}{2} M \Omega^2 \hat{X}_i^2 \right) + \alpha \sum_{\sigma,i} \hat{X}_i \left( \hat{n}_{\sigma,i} - \frac{1}{2} \right).$$
(15)

Here, $c^\dagger_{i,\sigma}$ ($c_{i,\sigma}$) creates (annihilates) a spin-$\sigma$ ($=\uparrow, \downarrow$) electron at site $i$, and $\hat{n}_{\sigma,i} = c^\dagger_{i,\sigma} c_{i,\sigma}$ is the corresponding number operator. The nearest-neighbor hopping integral is t, $\mu$ is the chemical potential, and $a$ is the lattice constant. A phonon mode with energy $\Omega$ and mass $M$ is placed on each site $i$, with $\hat{X}_i$ and $\hat{P}_i$ the corresponding position and momentum operators, with $\alpha$ the e-ph coupling strength. In our tests, we set $t = M = a = \Omega = 1$ and $\alpha = \sqrt{2}$, and consider a half-filled $\langle n \rangle = 1.0$ ($\mu = 0$) system. The corresponding dimensionless coupling is $\lambda = \frac{\alpha^2}{MW\Omega^2} = 0.5$, where $W = 4t$ is the bandwidth of the non-interacting model. For these parameters, the model has an insulating $q = \pi/a$ charge-density-wave (CDW) ground

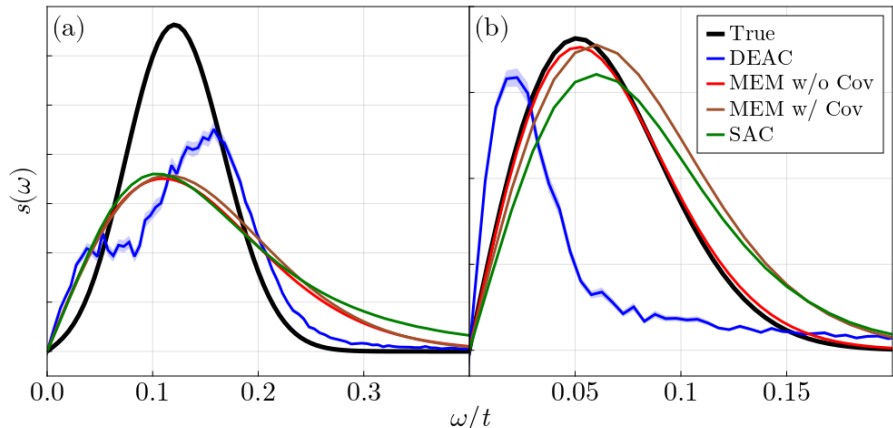

Figure 4: Bosonic cases where DEAC underperforms. (a) $s(\omega)$ is a Gaussian centered at $\mu = 0.1$, width $\sigma = 0.05$. (b) $s(\omega)$ is a Gaussian $\mu = 0.01$, width $\sigma = 0.05$.

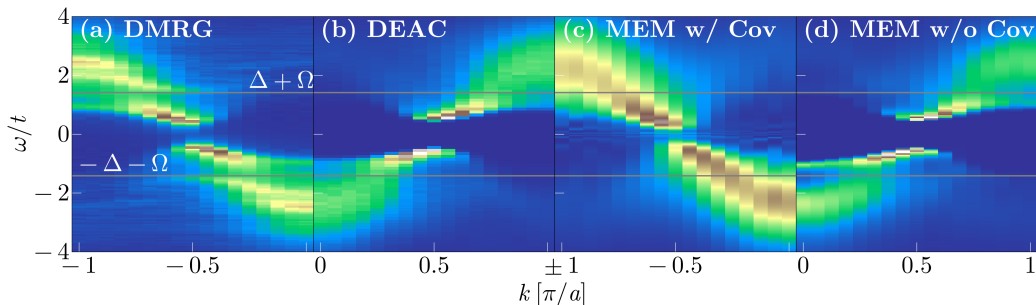

Figure 5: Results for the fermionic spectral function $\mathcal{A}(k, \omega)$ of the half-filled Holstein model in 1D. (a) DMRG results at $T = 0$. (b)-(d) DQMC results at $T = 1/\beta = 0.025t$, analytically continued using the DEAC and MEM methods with and without the covariance method, as indicated. The model parameters for all simulations are $\Omega = t$ and $\alpha = \sqrt{2}$. The gray lines indicate the energy $\pm(\Delta + \Omega)$, where $\Delta$ is the magnitude of the gap measured in the DMRG simulations at $k = \pm\pi/(2a)$.

state [30].

The DQMC simulations were performed using the `SmoQyDQMC.jl` package [31,32] for a fixed inverse temperature $\beta = 1/T = 40/t$ and $\Delta\tau = 0.05/t$. We performed 2,500 burn-in and 5,000 measurement sweeps across 10 MPI ranks to generate a total of 100 data bins for each measured quantity. The electron spectral function is obtained from the time-displaced Green's function $G_\sigma(k, \tau) = \langle T_\tau c_{k,\sigma}(\tau) c_{k,\sigma}^\dagger(0)\rangle$. Similarly, the phonon spectral function is obtained from the phonon Green's function $D(\mathbf{q}, \tau) = 2\Omega M \langle T_\tau X_q(\tau) X_q(0)\rangle$. We then used `SmoQyDEAC.jl` and both MEM implementations to obtain the resulting electron $[\mathcal{A}_\sigma(k, \omega)]$ and phonon $[\mathcal{B}(q, \omega)]$ spectral functions using the kernels defined in Eq. (6). The DMRG results were obtained at zero temperature using the DMRG++ [33,34] code, keeping $m = 500$ states to maintain a truncation error below $10^{-7}$. We restricted the local phonon Hilbert space to keep $N_p = 20$ phonon quanta per site and checked that our results converged for both the number of phonon modes and DMRG states. The spectral functions were computed directly on the real frequency axis using the correction-vector algorithm with Krylov-space decomposition and a one-site update [35]. The electron spectral function was calculated from the sum of the

electron addition $[\mathcal{A}_{ij}^+(\omega)]$ and removal $[\mathcal{A}_{ij}^-(\omega)]$ spectra. In real space, they are defined as

$$\mathcal{A}_{ij}^{\pm}(\omega) = -\frac{1}{\pi} \operatorname{Im} \langle \Psi_{\mathrm{gs}} | \hat{A}_j \frac{1}{\omega - \hat{H} + E_{\mathrm{gs}} + \mathrm{i}\eta} \hat{B}_i | \Psi_{\mathrm{gs}} \rangle , \tag{16}$$

where $|\Psi_{\mathrm{gs}}\rangle$ and $E_{\mathrm{gs}}$ are the ground state and corresponding energy, respectively, with operators $\hat{A}_j = c_{j,\sigma}$, $\hat{B}_i = c_{i,\sigma}^\dagger$ ($\hat{A}_j = c_{j,\sigma}^\dagger$, $\hat{B}_i = c_{i,\sigma}$) for the addition [+] (removal [−]) spectra. The final spectral function in momentum space $\mathcal{A}(k,\omega)$ was then obtained by Fourier transforming Eq. (16). In our calculations, we fixed the broadening coefficient to $\eta = 0.1t$ and shifted spectra by the chemical potential $\mu = (E_{\mathrm{gs}}^{N+1} - E_{\mathrm{gs}}^{N-1})/2$, where $E_{\mathrm{gs}}^N$ is the ground state energy of the system with $N$ particles, to place the Fermi energy at $\omega = 0$.

Figure 5 compares electron spectral functions obtained from the four methods, with the DMRG spectra at $T = 0$ shown in panel (a) and the DQMC spectra at $T = 0.025t$ in the three subsequent panels (b)-(d). The DMRG results show a gap $\Delta$ due to the CDW ground state and exhibit the typical band renormalizations associated with a strong $e$-ph interaction [4,36, 37]. These include the Engelsberg-Schrieffer [38] kink at $\pm(\Delta + \Omega)$ and a secondary two-phonon feature at $\pm(\Delta + 2\Omega)$. All three of the DQMC-derived spectra capture the presence of the gap; however, the DEAC and non-covariance MEM methods slightly overestimate its size, while the covariance matrix/Bryan algorithm based MEM underestimates it. Both DEAC and non-covariance MEM methods also capture many aspects of the renormalized band structure, including the dispersion kink at $E = \pm(\Delta + \Omega)$ and its trailing intensity as the momentum tracks toward $k = 0$. Conversely, the covariance matrix/Bryan algorithm based MEM approach produces a very broad spectrum and is unable to resolve the dispersion kinks. All three AC methods fail to capture the two-phonon feature visible in the DMRG results.

Comparing the sharpness of the spectral features, we observe that the DMRG results are broader than the DQMC results for $|\omega| \leq \Delta + \Omega$, while the reverse is true at higher energies. We attribute this behavior to the finite broadening $\eta = 0.1t$ used in the DMRG simulations. In general, the imaginary part of the self-energy in this case will be very small at low temperatures for $|\omega| \leq \Delta + \Omega$ due to restrictions in the phase space for scattering [39]. The line width of the DMRG spectra is thus set by $\eta$ in this energy window, resulting in broader spectral features than would otherwise be expected.

Figure 6 provides a similar comparison for the phonon spectral functions $\mathcal{B}(q, \omega)$. As the model has strong $q = \pi/a$ CDW correlations at low temperatures, we do not include this momentum in panels (a)-(c) as it would overwhelm the signal throughout the remainder of the Brillouin zone. The spectral functions at the zone boundary are instead plotted separately in Fig. 6(d).

The DMRG results are strongly renormalized and exhibit a two-peak structure similar to those observed in previous QMC simulations [40]. Specifically, a portion of the spectra softens at the zone boundary due to the $q = \pi/a$ CDW correlations in the system, while a remnant of the original phonon dispersion persists near the zone center. (We attribute oscillations in intensity near zero energy to finite size effects in the Fourier transform.) Both the DEAC and MEM results without covariance reproduce the softening and overall dispersion of the renormalized phonon branch. However, they also have suppressed spectral weight near $q = \pi/4a$, where the DMRG results have kink-like break in intensity. To further validate these results, we have also calculated the renormalized phonon frequencies using the relationship [41]

$$\Omega(q) = \sqrt{\Omega_0^2 + \Pi(q,0)} = 1/\sqrt{D(q,0)}, \tag{17}$$

where $D(q,0) \equiv D(q, \mathrm{i}\nu_n = 0)$ is the the phonon Green's function. The calculated dispersion is overlaid with the DEAC and MEM results, and agree well with the dispersion of the spectra. Both methods are thus capable of accurately capturing general dispersion of the renormalized phonon branch throughout the first Brillouin zone.

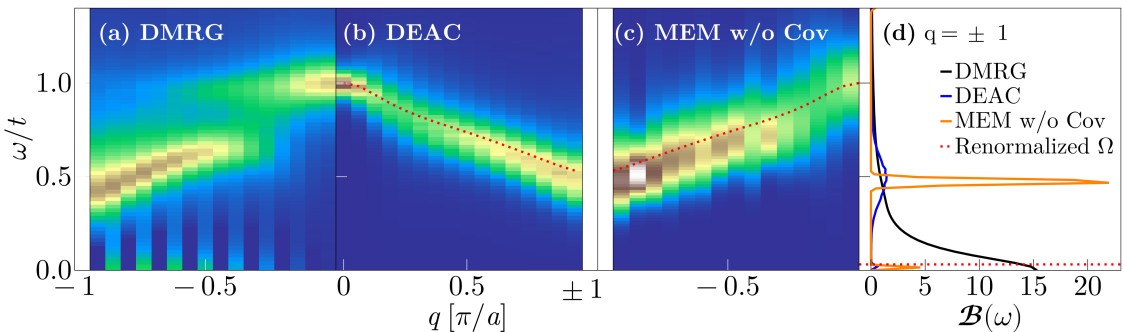

Figure 6: Results for the phonon spectral function $\mathcal{B}(q, \omega)$ for the half-filled Holstein model in 1D. (a) DMRG results at $T = 0$. (b) DQMC results at $T = 1/\beta = 0.025t$ analytically continued using the DEAC algorithm. (c) DQMC results for non-covariance MaxEnt. (d) Results for $q = \pm\pi/a$ plotted. The model parameters for both simulations are $t = \Omega = \langle n \rangle = 1$, and $\alpha = \sqrt{2}$.

Turning to the soft mode at $q = \pi$, shown in Fig. 6(d), we see that the DMRG result concentrates most of the spectral weight at $\omega = 0$, consistent with a ground state CDW. In contrast, both DEAC and MEM without the covariance matrix produce two peaks at $\omega = 0$ and $\omega = 0.5t$. In this case, DEAC broadens the higher energy peak and places little weight at zero energy while MEM produces two sharp peaks with more spectral weight at low energies. Both methods fail to capture the phonon spectral function at this wave vector.

## 4.6 Application to the 1D Hubbard-Holstein model

As our final benchmark, we also compared the electron and phonon spectral functions obtained using these methods for a $L = 32$ site Hubbard-Holstein chain. Its Hamiltonian is

$$\hat{H} = U \sum_i \left( \hat{n}_{\uparrow,i} - \frac{1}{2} \right)\left( \hat{n}_{\downarrow,i} - \frac{1}{2} \right) + \hat{H}_{\text{Hol.}}, \tag{18}$$

where $\hat{H}_{\text{Hol.}}$ is the Holstein Hamiltonian given in Eq. (15) and $U$ is the on-site Coulomb repulsion. In what follows, we keep the same Holstein parameters used in Sec. 4.5 and set the Hubbard repulsion to $U = 8t$. The model's ground state for these parameters is a Mott-Hubbard insulator [42]. We performed DQMC simulations at an inverse temperature of $\beta = 70/t$ and $\Delta\tau = 0.1/t$. We employed 5,000 burn-in and measurement sweeps each with 50 MPI ranks to generate 500 data bins. Our DMRG simulations kept $m = 500$ states and restricted the local phonon Hilbert space to keep $N_p = 15$ phonon quanta per site.

Figure 7 compares electron spectral functions obtained from three methods, with the DMRG spectra at $T = 0$ on the left and the $T = t/70$ DQMC spectra obtained with DEAC and MEM in the two subsequent panels. All spectra exhibit a robust gap at the Fermi level, as expected for its Mott-insulating ground state [42]. DMRG finds a gap width of $\delta = 2.1t$, whereas our DEAC and MEM runs found $\Delta = 2.4t$ and $\Delta = 2.15t$, respectively. The DMRG spectrum also exhibits several quasi-particle branches for $|\omega/t| \geq 2$, which are footprints of spin-charge separation. Neither DEAC nor MEM can reproduce these features; both methods smear the branches into a single broad peak, with MEM performing worse in this regard.

Figure 8 compares the calculated phonon spectral functions. The width of the DMRG result is almost entirely determined by our broadening parameter $\eta = 0.1t$, indicating that the phonon branch is very weakly renormalized deep in the Mott insulating regime. All three methods generally agree with this picture and produce a sharp, nearly dispersionless phonon peak with only a slight softening toward the zone boundary. Notably, MEM produces a peak

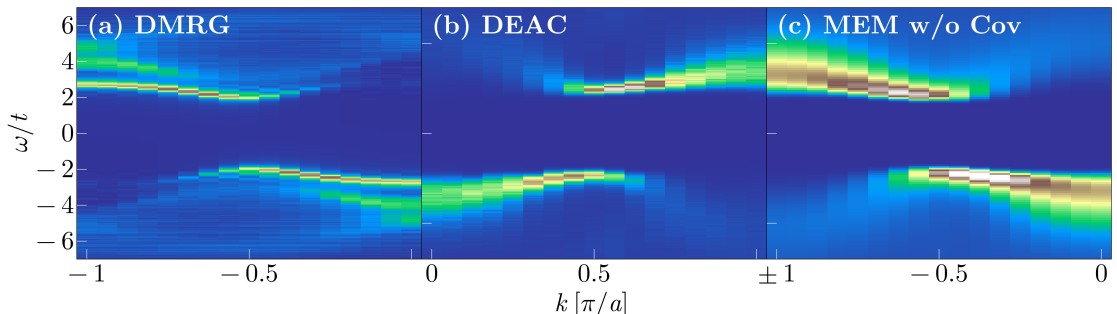

Figure 7: Results for the fermionic spectral function $\mathcal{A}(\mathbf{k}, \omega)$ for the half-filled Hubbard-Holstein model in 1D. (a) DMRG results at $T = 0$, (b) DQMC results at $T = 1/\beta = t/70$ analytically continued using the DEAC algorithm. The model parameters for both simulations are $t = \Omega = \langle n \rangle = 1$, and $\alpha = \sqrt{2}$. (c) DQMC results with MEM without covariance method.

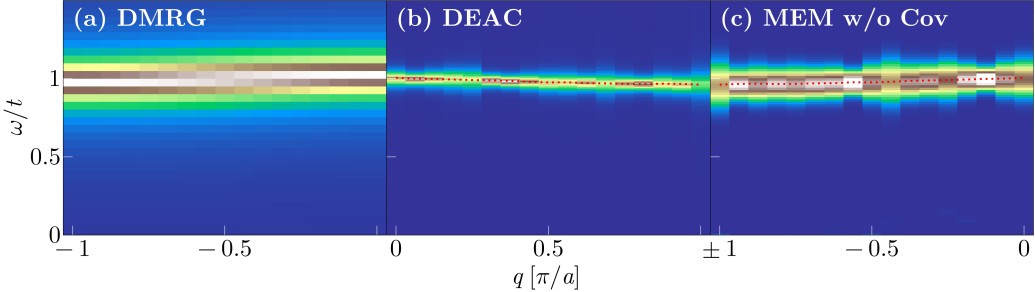

Figure 8: Results for the phonon spectral function $\mathcal{B}(q, \omega)$ for the half-filled Hubbard-Holstein model in 1D. (a) DMRG results at $T = 0$, (b) DQMC results at $T = 1/\beta = t/70$ analytically continued using the DEAC algorithm. The model parameters for both simulations are $t = \Omega = \langle n \rangle = 1$, $U = 8$, and $\alpha = \sqrt{2}$. (c) DQMC results with MEM without covariance method. The red dotted line represents renormalized phonon frequency calculated using Eq. (17).

with a width compatible with the DMRG result while DEAC produces a much sharper spectral function that reflects the actual (minimal) self-energy broadening of the phonons.

## 5 Concluding Remarks

We have introduced the `SmoQyDEAC.jl` Julia package for performing analytic continuation on noisy correlation function data generated with QMC methods. This package extends the capabilities of the previous implementation [19] by providing support for both fermionic and bosonic correlation functions defined on the either imaginary time $\tau$ or Matsubara frequency $\omega_n$ axis. It also utilizes a simple scripting interface, parallelization, and a fitness floor finder tool to enable fully automated analytic continuation.

While many methods exist for analytically continuing noisy correlation functions generated by QMC methods, they vary significantly in their ability to accurately replicate spectra. As of yet, there is no analytic continuation method that consistently outperforms all others in every situation. In this spirit, we have presented representative cases showing both when our differential evolution for analytic continuation implementation may be the best option, and when it is known to perform poorly. We have found that our implementation produces fairly reliable spectral functions from noisy QMC data when applied to fermionic correlation functions or bosonic correlation functions with relatively high-energy peaks.

Finally, we note that all of our test cases were run without tweaking any of the default settings for each implementation and results could likely be improved on with ad hoc adjustments. However, our intent is not to present a robust comparison of DEAC against other AC methods but give a representative overview of DEAC's performance out of the box without need for user intervention.

## Acknowledgements

**Funding information**: This work was supported by the U.S. Department of Energy, Office of Science, Office of Basic Energy Sciences, under Award Number DE-SC0022311. N.S.N. was supported by the Argonne Leadership Computing Facility, which is a U.S. Department of Energy Office of Science User Facility operated under contract DE-AC02-06CH11357.

## Code availability

The code used to generate our benchmark tests, as well as additional benchmark figures, are available at https://zenodo.org/records/10407525.

## A Noise reduction with increasing numbers of genomes

Each DEAC-generated genome produces a noisy spectrum and, generally, many genomes are averaged to obtain a smoother spectrum. To illustrate how the noise levels decrease with the number of genomes, we generated bins of 100 genomes for the same test case shown in Fig. 2e. Figure 9 then plots the results of averaging over 1, 10, 40, and 100 of these bins to obtain the final spectral function. In this case, we find that the noise levels decrease significantly with additional genomes, but the rate of reduction decreases as the number total number of genomes increases.

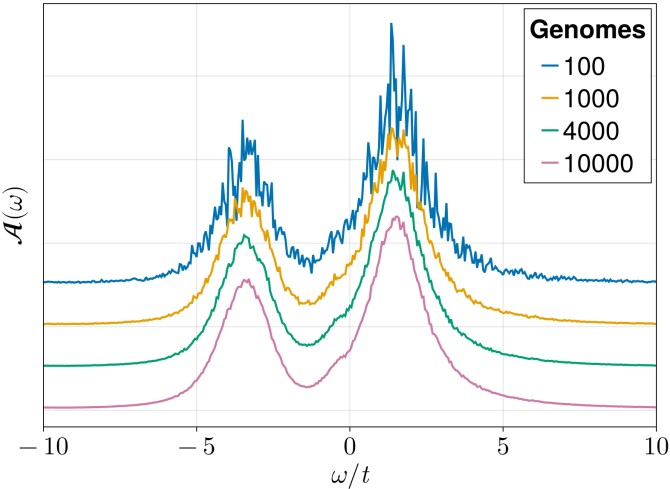

Figure 9: An example of how the noise level decreases as one averages over more genomes. The test case used here is identical to the one used in Fig. 2e.

## B  FFF testing

We have found `SmoQyDEAC.jl`'s built in fitness finder does a reliable job finding a fitness target below which times-to-solution become significantly burdensome. As we show in Fig. 10, the FFF will generally find a fitness typically obtainable within $100,000$ generations, with only 5 of 48 threads not converging within that number of generations for our plotted case. However, their fitness was not far off of the the value found by the FFF (dotted red line). We stress that the FFF's value for $\chi^2$ should not be construed as an idealized value for the fitness parameter.

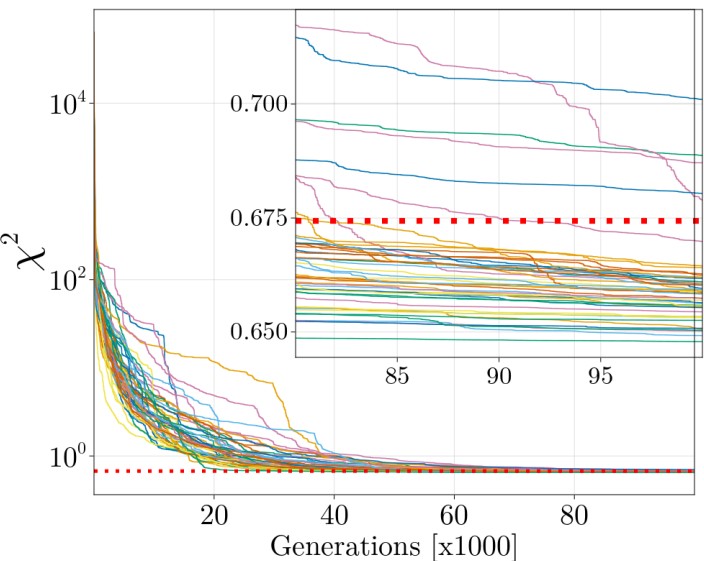

Figure 10: Genome fitness for 48 threads over 100,000 generations. Inset is last $20,000$ generations. Dotted red line is fitness found using our fitness floor finder (FFF).

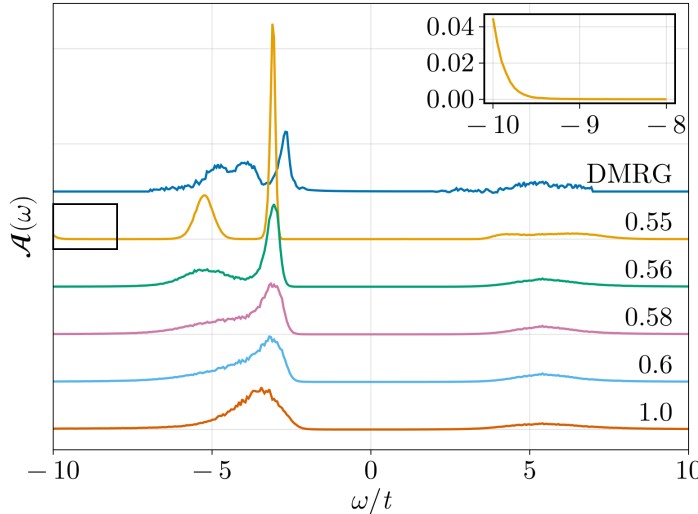

Figure 11: Hubbard Holstein spectral function for $k = 0$ plotted at various $\chi^2$ fitnesses and the DMRG result. Inset: characteristic overfitting tail for $\chi^2 = 0.55$

## C  Remarks on overfitting

Due to the existence of noisy inputs and the ill-posed nature of AC, it is possible for a highly divergent solution to have a lower $\chi^2$ fit value than a far more likely distribution. Using the DQMC $k = 0$ electronic Green's functions from the Hubbard-Holstein example (Fig. 7), Fig. 11 plots a series of DEAC results obtained for different selected target fitnesses. Here, we find that the DEAC result converges towards the DMRG result as we go below a fitness target of $\chi^2 = 1$; however, the results fairly rapidly transition into an overfitting regime at a target of $\chi^2 = 0.55$. In this case, DEAC finds two relatively sharp peaks that do not correctly reproduce the DMRG result. Notably, we also see a pile-up of spectral weight at the boundary of our frequency range, leading to an upward-curving tail (see Fig. 11, inset). We have found that such a feature is often a sign of overfitting.

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
