# Peer review of "SmoQyDEAC.jl: A differential evolution package for the analytic continuation of imaginary time correlation functions"

_SciPost Physics Codebases, doi:SciPost Phys. Codebases 39-r1.1 (2024) , SciPost Phys. Codebases 39 (2024)_

## Round 1 · Referee Report · Anonymous (Referee 1) · 2024-9-14

Strengths

  1. clear documentation for API and examples
  2. well-formatted manuscript

Weaknesses

NA

Report

The authors present a Julia package that implements the Differential Evolution Analytic Continuation (DEAC) method, a technique recently proposed for continuing noisy correlation functions on the imaginary axis. This software package could benefit the condensed matter community, especially those lacking computational expertise necessary for performing analytical continuation on simulation data.

The manuscript is well-written and the documentation is quite clear.
The authors provide a handful benchmark against other well-established methods, which should help users become familiar with both the package and the underlying algorithm.

I only have one minor question before recommending this manuscript for publication in SciPost.

Requested changes

  1. The DEAC results often exhibit oscillating curves, as shown in Figures 2, 3, and 4, where the authors attribute these oscillations to noise from Monte Carlo (MC) sampling. However, I find it somewhat confusing to determine whether the oscillations stem from insufficient MC sampling or from overfitting. In other words, are all the oscillations in Figures 2, 3, and 4 suppressible as the number of MC sampling increases? Could the authors demonstrate the convergence with respect to the number of runs for one or more of the selected test systems? 

Such an analysis would be highly instructive, providing users with clear guidance on how to assess the quality of their DEAC results.

Recommendation

Ask for minor revision

  • validity: top
  • significance: high
  • originality: good
  • clarity: top
  • formatting: perfect
  • grammar: perfect

Author:  James Neuhaus  on 2024-10-02  [id 4823]

(in reply to Report 1 on 2024-09-14)
Category:
remark
answer to question

We thank the Reviewer for their time and for their support for publication.

We believe that the oscillations in the DEAC data referenced by the Reviewer are the rapid
ones shown in each case due to the statistical noise introduced by sampling the population. One can
systematically decrease this noise level by averaging over more genomes. To demonstrate this, we have
added a new appendix to the paper that shows the evolution of the predicted spectra shown in Fig. 2e
as the total number of genomes increases. We have also added another appendix that discusses some
common spectral features we have encountered when we overfit a given spectrum. Attached are the two new figures which are included in the new submission. A noise vs number of genomes plot is on the left hand side, and a plot of target fitness is on the right hand side. Full explanations are included in the two new appendices.

Thank you.

Attachment:

---

## Round 2 · List of Changes



---

## Editorial Decision

published